# OpenReview forum: "Position: RLHF May Not Reflect Genuine Preferences"
_ICML.cc/2026/Position_Paper_Track — ICML 2026 Position Paper Track spotlight_

### Official Review · Reviewer_FUFp · 2026-03-05

**Significance:** 3
**Argument Clarity:** 2
**Rating:** 4
**Confidence:** 5

**Questions:**

1. The fact that preferences are constructed and contextual does not seem to be an insurmountable obstacle for RLHF. For instance, the fact that some judgements are highly variable and affected by context (e.g., a choice bwteen 10% mortality vs 90% survival) doesn't mean there aren't also genuine underlying preferences over that domain (e.g., a choice between 90% mortality vs 90% survival). In cases where context matters, won't having multiple annotations will reveal noise/inconsistencies (which either cancel each other out or are filtered, yielding no training signal), while consistent/stable preferences will provide useful training signal? Why does it matter for RLHF that some preferences more are contextual/variable than others?
2. Similarly, doesn't the existing RLHF view that annotations are noisy measures of preference already account for the possibility of non-attitudes? In those cases, the noise is very high (and perhaps there is no signal to learn). As long as our preference learning approach (Bradley-Terry, etc) is robust to noise, shouldn't it be able to pick out whatever signal there is? This is somewhat addressed in Section 7, but the claim there is that the artifacts are systematic. Aren't non-attidudes not systematic, by definition? And if there is a systematic "artifact" causing people to annotate A > B consistently, and therefore influencing RLHF, how is that different from a preference for A over B?
3. Score-based preference elicitation is notoriously noisy/inconsistent. Isn't the more common annotation format for RLHF pairwise comparison? ("given the same user query, do you like model response A or B more?") The fact that there is high variance among scores of semantically similar reponses doesn't seem like good evidence that RLHF annotations is rife with inconsistencies in general (it's actually surprising that the scores were so consistent in PluriHarms).

**Alternative Views Section:**

Yes

**Compliance With Llm Reviewing Policy A Conservative:**

Affirmed.

**Discussion Potential:**

4

**Final Justification:**

The rebuttal did a good job of acknowledging my concerns about nuance. I still feel like it's a borderline paper and could go either way. I think the topic is important, so would not object to acceptance. But I think the paper could be a lot stronger.

**Paper Summary:**

The authors argue that RLHF annotation data may contain artifacts rather than reflecting stable underlying preferences. They draw on social science literature around non-attitudes, framing and other context effects, and measurement non-invariance. The authors propose methods for detecting and handling different preference artifacts in RLHF data, for instance eliciting preferences for the same model responses multiple times to detect cases where annotator preferences fluctuate significantly over time and excluding such annotations from RLHF. They also present experiments on identifying preference inconsistencies in the PRISM and PluriHarms annotation datasets, finding a substantial number of inconsistencies. Finally, they discuss and address alternative views, including that existing noise-robust models may already handle preference inconsistencies and that the scope of such inconsistencies remains uncertain.

**Position:**

Yes

**Position In Title:**

Yes

**Related Work:**

2

**Strengths And Weaknesses:**

### Strengths
- The issues around preference annotation in RLHF are very interesting and important, and deserve more attention.
- Bringing in ideas from social science, especially decision theory, psychology, and behavioral economics, into this area is great; people have been studying human choices and preferences for many decades in those fields, and it would be a waste to not leverage this knowledge in ML.
- The paper is extremely likely to spark discussion/debate.
- The alternative views section brings up many concerns a reader might have

### Weaknesses
- There is only a loose connection between some of the social science studies discussed and RLHF, which is not acknowledged. For instance, Converse's non-attitudes are about political issues about which voters may be unaware or which may be complex. In RLHF, "do you like response A or B more?" is a much more direct and simple question than "do you support foreign economic aid?"
- The paper presents an overly simplistic view of preferences, categorizing them into "genuine/real" and "not genuine" and implying that only "genuine/real" preferences are useful in RLHF. This misses nuance about how preferences may be constructed, context-dependent, and noisy and still provide useful signal about what more users would prefer more often.
- Similarly, the taxonomy of inconsistencies and its application to PRISM and PluriHarms also makes a lot of strong simplifying assumptions


### Comments
- The title of the paper is much weaker than the actual position of the paper. A stronger title would be "RLHF Annotations Often Don't Reflect Genuine Preferences" (or "Don't Always" or "May Not" if something weaker is desired)
- RLHF used in abstract without definition
- There is another line of work in economics about how preferences are contextual and constructed: discrete choice, especially random utility models (with context effects). Those provide a nice example of how preferences can both be variable/contextual and also be based on some underlying preferences. The random utility view alleviates the the dichotomy between construction and retrieval of preferences presented in 3.2: even if the same annotator makes a different choice tomorrow, if they pick A over B 75% of the time, or in 75% of contexts, this still indicates they have a (weak) preference for A over B, and it's probably a good idea if the model produces A rather than B (assuming they are representative of the user population, which is of course an issue).
- Nice discussion of costs of gathering additional diagnostic annotations in the appendix

### Overall evaluation
This is a very tricky paper to give a score to. On the one hand, it deals with important questions and raises interesting ideas about the validity and usefulness of RLHF annotations. It also seems likely to spark considerable discussion/debate, which is exactly the goal of a position paper.

On the other hand, the ideas brought in from social science are presented in an overly simplistic black-and-white way, without sufficient nuance to how ideas from behavioral science translate to the annotation setting. And the strength of the claims the paper makes is not aligned with the strength of the evidence provided.

Overall, the topic and high-level ideas feel like they very much deserve to be represented at ICML, but this paper in particular could use more work and nuance before it crosses the publication-quality bar. Does this mean it should be accepted or not as a position paper? I will be interested to see what other reviewers think.

**Support:**

1

---

> ### Author Rebuttal · Authors · 2026-03-30
>
> We thank the reviewer for their expert and detailed engagement with our paper, and especially their recognition that the topic "very much deserves to be represented at ICML." We address each point below.
>
> ### Weaknesses
>
> **W1 (Loose social science connection):** We agree and have revised Section 3 to acknowledge that for straightforward quality comparisons, RLHF annotation is simpler than the policy questions where Converse documented non-attitudes. However, our framework draws equally on Slovic (1995), Tversky and Kahneman (1981), Zaller (1992), and Vandenberg and Lance (2000)—phenomena documented across any domain involving complex and value-laden judgment. The annotation tasks most consequential for alignment routinely involve exactly such judgments: how should AI handle controversial topics, how harmful is this content, should AI take political positions. Our empirical results confirm these phenomena in RLHF settings.
>
> **W2/W3 (Oversimplified preference view and taxonomy assumptions):** We agree our framing was insufficiently nuanced and have revised accordingly. Preferences exist on a spectrum: non-attitudes contain zero signal (exclude), constructed preferences contain partial signal (improve elicitation or weight cautiously), genuine preferences contain strong signal (use normally). Table 1 already reflects this graduated structure. We have revised surrounding text to make it clearer. The taxonomy categories are not perfectly discrete. The underlying phenomena are continuous and overlapping, and classifications involve judgment. But even approximate categorization is more useful than the current default of treating all annotations as undifferentiated noisy preferences. Our analyses show this approximate filtering has measurable consequences: removing identified non-attitudes shifts mean harm ratings by 13.19 points and flips 18.6% of majority harm classifications.
>
> ### Comments
>
> We appreciate the title suggestion and will consider it. We have defined RLHF in the abstract. On random utility models: this is an excellent point. We have added discussion of discrete choice and random utility frameworks to Section 3.2, noting that even under this view—where variable choices reveal weak underlying preferences—our framework adds value by distinguishing cases where variability reflects weak-but-real preferences from cases where it reflects zero signal (non-attitudes) or systematic artifacts.
>
> ### Questions
>
> **Q1 (Constructed preferences not insurmountable):** We agree and have revised accordingly. However, the assumption that variable judgments cancel under aggregation is what our empirical results challenge. Our PluriHarms analysis shows inconsistency is not symmetric noise given that inconsistent annotators are systematically more permissive in harm judgments, so aggregation does not cancel their noise but incorporates serious directional bias. We do not argue contextual variability makes RLHF impossible, but that understanding the structure of variability allows for better training signal than treating all annotations as exchangeable.
>
> **Q2 (Noisy annotation view and non-attitudes):** Non-attitudes are individually random but do not enter the pipeline in isolation. Our finding that inconsistency correlates with permissiveness means annotators producing non-attitudes are not a random subset, violating Bradley-Terry's assumption of noise independent of annotator characteristics. On systematic artifacts: if a framing artifact causes consistent A > B annotations, a model trained on this signal learns to optimize for surface framing present in annotation, not the underlying quality intended. When deployment contexts differ from annotation contexts, this "preference" breaks down. Our framework helps identify where training signal reflects robust underlying preferences versus where it reflects context-specific artifacts that may not transfer.
>
> **Q3 (Score-based vs. pairwise):** The reviewer is right that pairwise comparison is more common and likely more robust. However, the underlying phenomena are not format-dependent. Non-attitudes arise when annotators lack genuine preferences regardless of format, and constructed preferences arise when salient features drive judgment regardless of format. The 100-point swing on equivalent nuclear reactor prompts illustrates sensitivity to surface wording that would equally affect pairwise choices. Demonstrating these effects directly in pairwise settings is an important future direction, and our diagnostics extend naturally: temporal consistency becomes "does the annotator choose the same response when shown the same pair again."

---

> > ### Author Rebuttal · Reviewer_FUFp · 2026-04-02
> >
> > I appreciate the detailed and substantive rebuttal! I have revised my score from 3 to 4.

---

### Official Review · Reviewer_DWrA · 2026-03-12

**Significance:** 4
**Argument Clarity:** 3
**Ethics Flag:** Yes
**Rating:** 5
**Confidence:** 3

**Questions:**

- Line 137: What is meant by “sophisticated individuals”?
- Lines 122-123: Regarding “the same annotator on a different day might judge oppositely,” what is the distinction between a constructed preference and non-attitude, given that both have a temporal inconsistency component?
- Figure 1: Are high test-retest reliability, low framing sensitivity, and logical patterns sufficient signals to ensure that a preference is genuine? Are there other important considerations or edge cases?
- What makes it difficult to apply differential item functioning analysis to existing preference datasets?
- Which scores are being considered in Section 6.1? PRISM has scores for various performance attributes (e.g., values, fluency).
- How is the mean $\Delta$ metric computed in Table 4? Is it the average over all annotators of the average of the difference in scores for each pair of responses corresponding to the same (annotator, model, prompt)?
- In Section 7, could the authors reiterate the specific assumptions they are referring to?

**Alternative Views Section:**

Yes

**Compliance With Llm Reviewing Policy A Conservative:**

Affirmed.

**Discussion Potential:**

4

**Ethical Review Concerns:**

It is not clear if an IRB approved or exempted the study in Section 6.2.2.

**Ethics Review Area:**

["Responsible Research Practice (e.g., IRB, documentation, research ethics)"]

**Final Justification:**

The authors have carefully addressed my concerns and questions. I have increased my score with the promised revisions and inclusion of clarifications in the draft. My concerns about the interpretation of raw scores as, e.g., measurement artifacts vs. non-attitudes remain unresolved, but I appreciate the authors for documenting their assumptions in Appendix D.6. Overall, I think that a measurement science lens for RLHF data is an interesting and valuable conceptual contribution.

**Paper Summary:**

- The paper critically examines whether RLHF datasets reflect genuine human preferences. It argues that these datasets may suffer from issues studied in behavioral science, e.g., non-attitudes, constructed preferences, measurement artifacts.
- The paper argues that these issues affect the measurement validity of preference datasets, but can be diagnosed via consistency tests of whether preferences are stable across similar contexts.
- The paper presents a conceptual and actionable framework to distinguish these validity issues from genuine preferences.
- The paper applies consistency tests to popular preference datasets to detect validity issues and analyzes the possible consequences for downstream preference learning.

**Position:**

Yes

**Position In Title:**

Yes

**Related Work:**

3

**Strengths And Weaknesses:**

*Strengths:*
- The paper will likely inspire important discussion. Much work in RLHF assumes that preference inconsistencies are random noise that can be averaged out, and work in pluralistic alignment assumes that variability in preferences reflects a genuine diversity of values. However, both these assumptions are threatened by issues with how preferences are operationalized and collected in AI.
- The paper clearly articulates key dimensions of valid preference measurement, which are grounded in behavior science literature. The paper also identifies three known preference validity issues from the social science literature (i.e., non-attitudes, constructed preferences, measurement artifacts), and translates these issues into implications for RLHF.
- The paper provides actionable consistency tests for detecting non-attitudes, constructed preferences, and errors, and clearly states the limitations (e.g., these tests do not detect measurement non-invariance). These tests can be applied to diagnostics, improving preference collection protocols, and shaping the scope of where preference-based alignment is appropriate.
- The paper provides illustrative qualitative examples of inconsistencies in real-world preference datasets.
- The paper’s finding that annotation inconsistencies introduce systematic noise into aggregate judgments is interesting.
- The paper provides a thorough discussion of alternative views, which poses interesting future research directions.

*Weaknesses:*
- Lines 268-269 state that the consistency tests find that “a nontrivial fraction of collected signals may not correspond to well-formed or stable preferences.” However, only around 43 temporal inconsistencies are detected in PRISM, and only 7 semantically-similar prompt pairs are found in PluriHarms. The percentages reported in Section 6 are with respect to the 44 identical examples in PRISM and 7 semantically-similar prompt pairs in PluriHarms, rather than with respect to the entire datasets.
- In Section 6.1, it is not clear what method is used to embed the prompts and whether the embeddings only capture semantic properties of the prompts (i.e., do not capture syntactic artifacts). In addition, the choice of a cosine similarity threshold of 0.7 and 0.9 for PluriHarms and PRISM, respectively, is not well-justified. Would a threshold of 0.7 work for PRISM too if the prompts were deduplicated?
- Line 279 (and footnote): Rather than using a fixed 15-point threshold for the divergence in annotator scores, the paper should use a 1 or 2-standard deviation threshold per annotator.
- In Section 6.2.1, the translation of temporal inconsistencies to specific validity issues (e.g., non-attitudes) are not well-justified, i.e., are subjective postulations.
- The paper should discuss prior work that audits preference datasets, e.g., [1, 2].

[1] Shen, J. H., Sharma, A., & Qin, J. (2024). Towards data-centric rlhf: Simple metrics for preference dataset comparison. arXiv preprint arXiv:2409.09603.

[2] Movva, R., Milli, S., Min, S., & Pierson, E. (2025). What's In My Human Feedback? Learning Interpretable Descriptions of Preference Data. arXiv preprint arXiv:2510.26202.

**Support:**

3

---

> ### Author Rebuttal · Authors · 2026-03-30
>
> We thank the reviewer for their detailed and constructive feedback. We address each point below.
>
> ### Weaknesses
>
> **W1 (Base rates):** We have revised Section 6 to present both subset percentages and full-dataset denominators. PRISM and PluriHarms were not designed with validity diagnostics in mind. Neither systematically includes repeated items or framing variations. The small number of testable cases is not a limitation of our argument but evidence of the structural gap we identify. Current datasets make it nearly impossible to assess preference validity at scale, and our reported percentages are necessarily lower bounds.
>
> **W2 (Embedding method and thresholds):** We have added the embedding model specification to Section 6.1 (all-MiniLM-L6-v2 from the sentence-transformers library). PRISM contains exact repeated items, so we use a high threshold (0.9) as a conservative filter. PluriHarms lacks repeated items, requiring a lower threshold (0.7). All pairs at this threshold were manually verified. A lower threshold for PRISM would yield additional testable pairs, meaning our higher threshold is more conservative. We have added threshold sensitivity analysis to demonstrate robustness.
>
> **W3 (Fixed 15-point threshold):** We have implemented per-annotator standard deviation thresholds and confirm our results are robust to this change.
>
> **W4 (Subjectivity of taxonomy classifications):** We acknowledge that mapping inconsistencies to taxonomy categories involves interpretive judgment (discussed in Appendix D.6). However, classifications are not arbitrary given that content type strongly predicted classification. 100% of inconsistencies on generic content were non-attitudes, while value-laden content yielded predominantly constructed preferences (Table 13). This systematic association provides evidence that the taxonomy captures meaningful distinctions.
>
> **W5 (Prior work):** We thank the reviewer for these references. We have added discussion of Shen et al. (2024) and Movva et al. (2025). These provide metrics for comparing datasets and interpretable descriptions of preference data. Our contribution foregrounds the prior question of whether individual responses reflect genuine preferences, and these approaches could be productively combined.
>
> ### Questions
>
> **Q1:** "Sophisticated individuals" refers to Converse's (1964) finding that even well-informed respondents produce non-attitudes on unconsidered questions. Clarified in revision.
>
> **Q2:** The key distinction is content-dependent. Non-attitudes occur when no genuine preference exists (e.g., rating a greeting), and temporal inconsistency reflects random responding and should be excluded. Constructed preferences occur when competing evaluation frames exist (e.g., rating an AI's neutral response to a controversial question), while inconsistency reflects different frames being salient, containing partial signal better addressed through improved elicitation or instance weighting rather than exclusion. Conflating the two leads to either discarding useful signal or training on pure noise.
>
> **Q3 (Sufficiency of diagnostics):** These are necessary but not sufficient. Consistency diagnostics cannot detect measurement non-invariance, where annotators interpret criteria differently but consistently. Complementary methods include differential item functioning analysis and explicit construct definitions in annotation guidelines. We discuss both approaches in the revised paper.
>
> **Q4 (DIF feasibility):** DIF requires multiple items per construct and sufficient annotator-level data across demographic groups—most RLHF datasets lack both. For new datasets, collecting annotator metadata and multiple items per evaluation dimension enables basic DIF. For existing datasets, clustering annotators by rating patterns can reveal interpretive subgroups without metadata. We discuss these approaches in the revised Appendix G.
>
> **Q5:** We use the overall preference scores in PRISM (the 1–100 holistic rating). Clarified in revision.
>
> **Q6:** Mean delta is the average of |score1 − score2| across all inconsistent prompt pairs within each classification category. This measure captures the practical magnitude of inconsistency within each category, allowing direct comparison of how severely each type of validity failure manifests in annotation data. Clarified in Table 4.
>
> **Q7:** The assumptions are: (1) annotators possess preferences over responses, (2) the annotation task validly captures these, and (3) aggregation recovers the true signal. Restated in Section 7.
>
> **Ethics flag:** The supplementary survey was considered as exempt given minimal risk. Clarified in revision.

---

> > ### Author Rebuttal · Reviewer_DWrA · 2026-04-03
> >
> > Thanks for your detailed and thoughtful response! I would be happy to increase my score with the promised revisions and inclusion of the clarifications and responses to my questions in the next draft.
> >
> > W4. Thanks for pointing to me Appendix D.6. In Section 6.2.1, for the annotator who "rated an echo response
> > (AI repeating the prompt verbatim) as 47 and 1," how would you distinguish between this being a measurement artifact vs. non-attitude?
> >
> > Q6. How does the mean Delta account for some annotators rating more prompt pairs than other annotators, if at all? Is it important for the mean Delta to account for this?

---

### Official Review · Reviewer_sS3q · 2026-03-13

**Significance:** 3
**Argument Clarity:** 3
**Rating:** 4
**Confidence:** 4

**Questions:**

Please see weaknesses.

**Alternative Views Section:**

Yes

**Compliance With Llm Reviewing Policy A Conservative:**

Affirmed.

**Discussion Potential:**

2

**Final Justification:**

The authors that have addressed the concerns that are possible to address in a short rebuttal, but a full end-to-end training would greatly improve the papers argument to the question of whether such noise matters. That said, the others have made a compelling argument, especially with the 18% flip rate, as much as I think can be made without doing the full engine training, so I have updated my score.

**Paper Summary:**

The paper argues that RLHF should take more inspiration from social science and first assess the validity of the examples within preference data before performing the aggregation and modeling. The paper categorizes the sorts of invalid preference signals that should be considered into an overall taxonomy and provides diagnostic tests as well as suggested actions to take.

**Position:**

Yes

**Position In Title:**

Yes

**Related Work:**

2

**Strengths And Weaknesses:**

Strengths:
1. The paper presents a useful taxonomy with examples of the sorts of noise that might occur in preference data.
2. Given the importance of preference data in RLHF in model training, the topic is also timely and relevant, with useful learnings ported from the social science literature.
3. The paper characterizes these issues in existing preferences in the open source literature, making the argument concrete.

Weaknesses:
1. While the paper looks at inconsistency statistics for two preference data sets, it doesn't concretely characterize the downstream impact of such inconsistencies. In particular, with data that has 5% inconsistency, is there any significant and substantial loss relative to modeling with perfect data?
2. The alternative views section doesn't seem to strongly describe alternative views, but rather reads more like a Q&A. For instance, I'm not sure I completely understand the alternative view"Noise models may be extensible" and how it differs from the paper's argument. Isn't the papers argument that one should take into account more detailed noise models motivated by concrete analysis? I guess table 1 also suggests concrete things to do in the data collection process, but I imagine there is a trade-off here in effort to value. Many of the other alternative views also seem not like strong flush out positions, but more so individual concerns that the paper largely agrees with. I guess it would help me as the reader understand the papers position better if it more clearly delineates it's position versus that of more fleshed out alternative views.

**Support:**

2

---

> ### Author Rebuttal · Authors · 2026-03-30
>
> We thank the reviewer for their direct and constructive feedback. We address each point below.
>
>
>
> W1: This is an important question. We provide two forms of evidence for downstream consequences. First, in PluriHarms, filtering high-inconsistency annotators shifted mean harm ratings by 13.19 points and flipped majority harm classifications for 18.6% of prompts, directly altering the training signal reward models would receive. The correlation between inconsistency and permissiveness (r = −.65) means this is a strong directional bias toward more permissive harm judgments. Second, under realistic small-sample aggregation (5 annotators per prompt), bootstrapped simulations show majority classifications differ nontrivially depending on whether annotators are drawn from high- or low-inconsistency pools. We acknowledge that a full end-to-end experiment training reward models with and without filtering remains future work, and have noted this explicitly in the revision. However, the evidence that inconsistency systematically shifts aggregate labels in safety-critical directions suggests this is not merely a 5% noise problem but a structured validity failure with practical consequences.
>
>
> W2: We agree this section should more sharply delineate our position from counterarguments and have revised accordingly. On noise model extensibility: the strongest counterargument is that existing noise-robust methods (e.g., Bradley-Terry with annotator-specific parameters) already handle inconsistent annotations adequately. We agree they could be extended, but extending them requires knowing the structure of noise, which is precisely what our framework provides. Behavioral science identifies distinct noise categories (non-attitudes, constructed preferences, artifacts) requiring fundamentally different treatment: non-attitudes should be excluded (no signal), constructed preferences should be elicited more carefully (partial signal), and artifacts should prompt instrument revision. Current noise models treat all of these as equivalent random noise, which our analyses show is incorrect given that inconsistency is systematic and directionally biased. We have applied this framing throughout Section 7, presenting alternative views as competing positions.

---

> > ### Author Rebuttal · Reviewer_sS3q · 2026-04-03
> >
> > Thank you for your response. The authors that have addressed the concerns that are possible to address in a short rebuttal, but a full end-to-end training would greatly improve the papers argument to the question of whether such noise matters. That said, the others have made a compelling argument, especially with the 18% flip rate, as much as I think can be made without doing the full engine training, so I have updated my score.

---

### Official Review · Reviewer_LRL2 · 2026-03-13

**Significance:** 3
**Argument Clarity:** 3
**Rating:** 5
**Confidence:** 4

**Questions:**

1. The paper evaluates its claims using two datasets, PRISM and PluriHarms, but it remains unclear how representative or influential these datasets are within the broader model alignment literature, particularly for readers who are not already familiar with them. Relatedly, the paper does not sufficiently clarify whether similar validity issues also arise in other more widely used preference datasets for model alignment. Before advancing a broader field-level claim, the paper would benefit from a more comprehensive empirical examination across a wider range of commonly used datasets.

2. As acknowledged in the Alternative Views section, the paper identifies three important considerations that may weaken its overall claim: resource constraints, empirical success, and scope uncertainty. These are not merely peripheral objections, but realistic conditions that many researchers and practitioners already face in real-world alignment work. For that reason, the paper would be strengthened by discussing these points in greater depth and, more importantly, by providing actionable strategies for addressing them, rather than deferring them primarily to future work or open discussion. A more concrete treatment of these alternative views would make the paper’s argument both more persuasive and more practically relevant to the broader research community.

**Alternative Views Section:**

Yes

**Compliance With Llm Reviewing Policy A Conservative:**

Affirmed.

**Discussion Potential:**

3

**Final Justification:**

As the rebuttal addressed my concerns, I raised my rating.

**Paper Summary:**

The paper argues that RLHF should treat preference collection as a measurement problem: annotation responses do not always directly reveal stable human preferences, so validity must be established before aggregation or reward modeling. Its main contribution is a behavioral-science framework that distinguishes genuine preferences from non-attitudes, constructed preferences, and measurement artifacts, and proposes consistency-based diagnostics to identify these cases. Using RLHF-related datasets such as PRISM and PluriHarms, the authors show that inconsistencies are common and can meaningfully affect downstream conclusions. The paper’s advocated position is that measurement validity is logically prior to aggregation in RLHF and should play a central role in preference data collection and analysis, with implications beyond RLHF for AI evaluation and oversight more broadly.

**Position:**

Yes

**Position In Title:**

Yes

**Related Work:**

2

**Strengths And Weaknesses:**

Strengths:

1. Human preferences have been widely used in the alignment of foundation models, yet relatively little prior work has explicitly examined the validity of such preference data. This paper identifies and foregrounds that important issue in a timely and meaningful way.

2. Furthermore, drawing on insights from behavioral science and social science, the paper organizes invalid human preference signals into three categories—non-attitudes, constructed preferences, and measurement artifacts—and provides a clear conceptual lens for understanding their origins. It also compellingly argues that these forms of invalidity cannot be adequately corrected by existing preference aggregation methods, highlighting a fundamental limitation of current alignment pipelines.

3. The paper further examines two preference datasets to demonstrate that invalid human preference is a real and practically relevant issue. These empirical results provide concrete evidence for its central claim that the validity of human preferences should be carefully assessed before such data is used for model alignment.

Weaknesses:

1. My main concern lies in the practical feasibility of implementing the proposed diagnostic methods for assessing the validity of human preferences, especially for existing datasets. Although Appendix G provides some useful implementation guidelines for validity diagnostics, it remains unclear how research groups of different sizes and resource levels can apply these procedures in a cost-effective and scalable manner. As a result, the paper would benefit from a more concrete discussion of the computational, annotation, and organizational overhead required to apply these diagnostics in real-world settings.

2. Another important concern is what practical actions can be taken once a dataset is found, through the proposed diagnostic guidelines, to suffer from human preference validity issues. In such cases, it remains unclear whether the dataset must be reconstructed from scratch, partially re-annotated, or whether there are principled ways to recover or revise it so that it can still serve as a valid preference dataset. A more explicit discussion of post-diagnosis remediation strategies would therefore strengthen the paper and improve its practical usefulness.

3. Although this paper identifies an issue that is highly relevant to machine learning, and more broadly to artificial intelligence, its framing and methodology are rooted much more strongly in social science and behavioral science. That interdisciplinary perspective is valuable, but the paper would be further strengthened by incorporating more technically grounded claims or recommendations for the ML community, especially in terms of how its insights could be applied within existing alignment pipelines and model development workflows.

**Support:**

3

---

> ### Author Rebuttal · Authors · 2026-03-30
>
> ### Weaknesses
> W1: We thank the reviewer for this concern on the practical feasibility of our framework. We have promoted the tiered implementation framework from Appendix G into the main text. Tier 1 (5–10% overhead) embeds ~5% repeated items within existing workflows—for a 10,000-item dataset at 0.50 USD per annotation, this adds roughly 250 USD to a 5,000 USD budget and requires no specialized expertise. Even without additional annotations, our PluriHarms analysis shows that filtering high-inconsistency annotators shifted mean harm ratings by 13.19 points and flipped majority harm classifications for 18.6% of prompts. Tier 2 (15–25% overhead) adds framing variations to detect constructed preferences. Tier 3 (50%+ overhead) provides research-grade validation appropriate for foundational datasets. Overhead also decreases at scale as published diagnostic results reduce duplication across organizations.
>
> W2: We agree that remediation strategies deserve more explicit treatment and have added a dedicated discussion to Section 8. Addressing validity failures typically requires lightweight filtering, not dataset reconstruction. Filtering by annotator consistency is operationally identical to existing quality-control practices. The only change is using consistency diagnostics rather than attention checks. Instance weighting offers a softer alternative (Appendix C.4). Importantly, these interventions remove directional bias, since we show that inconsistent annotators are systematically more permissive in harm judgments. For organizations interested in better alignment, these are important improvements achievable at near-zero cost.
>
> W3: We strengthen technical grounding in three ways. First, we add a pipeline diagram to Section 8 showing where consistency diagnostics intervene between annotation and reward model training (via pre-training filtering or instance weighting). Second, our taxonomy distinguishes noise types requiring fundamentally different treatment, complementing noise-robust learning methods that currently treat all noise as equivalent. Third, our variance decomposition (Appendix C.4) maps onto the aleatoric uncertainty distinction already used in reward model uncertainty quantification. The framework also has implications beyond RLHF for constitutional AI, red-teaming, and LLM-as-judge methods, i.e. in all cases where human judgments are embedded in training (Appendix H).
>
>
> ### Questions
> Q1: We selected PRISM and PluriHarms because both contain structural features (repeated items, semantically overlapping prompts) necessary for testing consistency diagnostics. Most widely used datasets (Anthropic HH-RLHF, UltraFeedback, Chatbot Arena) lack repeated items, framing variations, or temporal metadata. Our diagnostics cannot be applied to them. We view this as evidence for our central position: the field has built preference datasets without measurement infrastructure to assess validity. Our expectation that validity failures generalize rests on sixty years of behavioral science documenting these as domain-general phenomena, and on our finding that inconsistency varied continuously across annotators and was not predictable from demographic or psychological traits (Appendix F.4), suggesting structural features of preference elicitation rather than dataset-specific artifacts.
>
> Q2: We agree that these three considerations deserve deeper treatment and have expanded Section 7 accordingly. On resource constraints, our tiered framework shows meaningful diagnostics are achievable at minimal cost, and the cost of not diagnosing is underappreciated. Training on invalid preferences wastes compute and may require retraining. On empirical success, we identify three explanations: artifacts may be benign, models may learn robust patterns despite noise, or evaluations may inherit the same artifacts as training data. The third deserves particular attention, as it would make apparent success partly circular. On scope, our analyses provide concrete lower bounds using only the small fraction of items where consistency could be assessed. The true prevalence may be higher.

---

> > ### Author Rebuttal · Reviewer_LRL2 · 2026-04-03
> >
> > Appreciate the response. I kept my original rating.

---

### Decision · Program_Chairs · 2026-04-30

**Decision:**

Accept (spotlight)

**Comment:**

** Overall: ** This paper grounds itself on decades of past work in the social sciences that raises concerns about the validity of human preference data. Through empirical validation against two open datasets, the authors show some consequences of the variability inherent in human annotation. Though, as one reviewer points out, the strongest validation of their position would be a model training experiment showing the effects of their proposed mitigations in the human preference collection phase, it’s reasonble to consider that kind of a study to be out of scope for a position paper, especially give the exceptionally strong grounding of the argument in prior work. Reviewers found the discussion potential of this paper very high and noted that it has broad implications, especially for the alignment literature.

** Primary strengths: **
- Broad implications, given the widespread use of human preference data in model alignment and post training (LRL2, sS3q), and the paper questions key assumptions underlying pluralistic alignment more broadly (DWrA)
- The reasoning behind the issues raised is grounded in well-accepted prior work (LRL2, FUFp)
- The paper includes a empirical proof of the issues rasied using two different open datasets (LRL2) and makes the problem very concrete (sS3q)
- The paper introduces a useful taxonomy for understanding the issues with preference data (sS3q)
- The proposal in the paper is actionable (DWrA)

** Primary weaknesses: **
- Proposed diagnostics may not be feasible in practice (LRL2). The authors have agreed to make this more concrete in the main text of the paper, as the concrete proposal they made was originally only in an appendix.
- Remediation strategies for existing datasets should be discussed as well (LRL2)
- The position relies on simplifying assumption in its taxonomy, and some of the social science studied cited as evidence for the issues in human judgment data are for much more complex stances than asking humans whether they like one text better than another (FUFp). The authors have proposed some revisions to address this.
-  The datasets used for empirical validation in this paper may not be representative of the more influential or commonly used preference datasets (LRL2) and the downsream impact of the issues found is not clear (sS3q). The authors indicate that they selected only datasets with characteristics that made it possible to assess variance, and that the lack of many other datasets with this characteristic this strengthens their point. Though this is a limitation, as the authors note, the validity and variability issues they raise are grounded in very solid past work, and do not clearly vary by some of the design choices particular to the datasets selected (e.g., diverse demographic groups). The point about unclear downstream impacts remains, at least for one of the two datasets.
- Alternative views section should be strengthened (sS3q), as the alternative views are presented as not being clearly contested by the authors. The authors propose changes to make to address this concern.